# Relationships between Fish Community Structure and Environmental Factors in the Nearshore Waters of Hainan Island, South China

Zhengli Luo [1,2], Changping Yang [1,3], Liangming Wang [1,3], Yan Liu [1,3], Binbin Shan [1,3], Manting Liu [1], Cheng Chen [1,3], Tao Guo [4,*] and Dianrong Sun [1,3,*]

1   South China Sea Fisheries Research Institute, Chinese Academy of Fisheries Sciences, Guangzhou 510300, China; luohua0303@163.com (Z.L.); yangchangping@scsfri.ac.cn (C.Y.); wangliangming@scsfri.ac.cn (L.W.); liuyan@scsfri.ac.cn (Y.L.); shanbinbin@scsfri.ac.cn (B.S.); liumanting@scsfri.ac.cn (M.L.); chencheng@scsfri.ac.cn (C.C.)
2   School of Fisheries, Zhejiang Ocean University, Zhoushan 316022, China
3   Key Laboratory for Marine Ranching, Ministry of Agriculture and Rural Affairs, Guangzhou 510300, China
4   Hainan Administration of Off-Season Plant Breeding and Seed Production Services, Sanya 572000, China
*   Correspondence: gtfree@126.com (T.G.); drsun73@scsfri.ac.cn (D.S.)

**Abstract:** The nearshore ecosystem of Hainan Island plays a crucial role as a breeding habitat for a wide variety of economically valuable fish species. Gaining insight into the structure of the fish community and the environmental factors that may impact them is highly significant in this marine region. This study collected fish data from 50 sampling sites using bottom trawling surveys. Methods including the swept area method, ecological indices, and abundance/biomass curve (ABC) analysis were employed to assess fish resource density and diversity. A comprehensive identification revealed a total of 363 fish species, encompassing 24 orders, 114 families, and 226 genera, with Perciformes being the dominant group. Spring exhibited higher fish abundance and biomass compared with autumn, with the southwestern waters as the primary concentration area. *Acropoma japonicum*, *Decapterus maruadsi*, and *Navodon xanthopterus* were dominant in spring, while *Leiognathus bindus*, *Saurida tumbil*, and *Champsodon atridorsalis* were dominant in autumn, indicating a seasonal shift towards smaller and lower-value fish species. A variability exceeding 80% was observed through SIMPER analysis, and a disrupted community structure was evident in the eastern and southern waters. Temperature and salinity were identified as primary environmental factors influencing the fish community. Overall, this study provides valuable insights into the nearshore fish community of Hainan Island, aiding in the understanding of its structure and dynamics.

**Keywords:** Hainan Island; fish community; environmental factor; seasonal changes; change in dominant species; canonical correspondence analysis





## 1. Introduction

Currently, the loss of biodiversity is a global trend. Fishing activities in the world have expanded from nearshore waters to offshore areas due to increasing disturbances, such as fishing pressure [1], the introduction of invasive species [2], and climate and environmental changes [3]. This expansion has led to a decrease in demersal fish catches in nearshore waters, consequently impacting fish community composition, ecosystem stability, and biodiversity. The protection and management of important fish communities in nearshore waters are crucial for maintaining sustainable fishery resources, the health of marine ecosystems, and global biodiversity.

Examining fish community composition and species diversity has been widely used to monitor environmental status and is a pivotal indicator for assessing fishery resources [4]. As a result, this field has garnered considerable attention from scholars. For instance, according to Kamrani et al. [5], environmental factors are the primary variables influencing

the species community structure in estuarine systems. Jorgensen et al. [6] demonstrated that seasonal variations in the marine environment exert a certain degree of influence on the succession of fish communities. Meanwhile, some studies have investigated the biodiversity, changes in dominant species, ecological niches, and interspecific relationships of fish communities in various marine regions, including the Mediterranean Sea [7], the northern slope of the South China Sea [8], and polar oceans [9].

Situated within the subtropical and tropical regions, Hainan Island stands as China's second-largest island. Hainan Island and its surrounding waters are recognized for their bountiful fishery resources, with fishing playing a pivotal role in the region's economy. While the overall number of fishing vessels has been declining since 2017, it continues to surpass the carrying capacity of offshore fishery resources. Moreover, the region also engages in widespread aquaculture practices involving feed, medication, and various aquaculture facilities. These activities have discernible impacts on water quality and the ecological environment [10]. Concurrently, Hainan Island flourishes as an immensely popular tourist destination, attracting a significant influx of visitors. The expansion of tourism may have multifaceted implications for the coastal marine environment, encompassing changes in land utilization, amplified pollutant discharges, and the diminishment of biodiversity [11,12]. However, the influence of temporal fluctuations in environmental variables on the utilization of fish communities in the nearshore waters of Hainan Island remains unclear. Understanding the response of nearshore fish to environmental changes enhances our understanding of the biology of nearshore fish and helps us comprehend the potential impacts of human activities on fish.

Hainan Island is widely recognized as a biodiversity hotspot in China, with extensive studies focusing on terrestrial plants [13], birds [14], and mammals [15]. Previous studies on aquatic organisms have primarily centered around freshwater fishes. Recent findings demonstrate a notable decline in both biodiversity and the general well-being of freshwater fish communities within Hainan Island [16]. Research specifically pertaining to the nearshore waters of Hainan Island is limited. Furthermore, investigations regarding the spatial and temporal distribution of the fish community structure and its connection to environmental factors in this marine region are scarce.

By using bottom trawl survey data collected in May and September 2022 from the nearshore waters of Hainan Island, this study examines the composition of dominant fish species, biodiversity, community structure, and their relationships with environmental factors. This work aimed to (1) enhance the existing data on fish resources surrounding Hainan Island and create an updated biological inventory of fish in the coastal waters; (2) thoroughly investigate the impacts of environmental changes on the fish community structure; and (3) provide a scientific foundation for the conversion, utilization, and sustainable development of marine fisheries in the area.

## 2. Materials and Methods

### 2.1. Study Area

Hainan Island is located in the northern of part of the South China Sea (108°37′00″–111°03′00″ E and 18°10′00″–20°10′00″ N). It is bordered by the Leizhou Peninsula across the Qiongzhou Strait to the north and shares borders with Qinzhou and Vietnam across the Beibu Gulf to the west. Hainan Island experiences a subtropical monsoon climate characterized by year-round warmth and humidity. The coastal waters of Hainan Island are influenced by the Beibu Gulf circulation, the western boundary current of the South China Sea, and the Qiongdong upwelling. Significant temperature and salinity variations exist within the biogeographic region, particularly between the eastern and western waters as well as the Qiongzhou Strait [17]. The island's unique geographical location and the complexity of its surrounding hydrological conditions make its nearshore fishing grounds one of the most significant in the South China Sea.

Moreover, Hainan Island serves as a critical spawning ground, foraging area, and nursery habitat for marine organisms, boasting abundant marine biological resources [18–20].

The nearshore waters of Hainan Island primarily receive river discharge originating from the island itself. The average annual rainfall on the island is 1794.12 mm [21], and the seasonal pattern of river flow resembles that of terrestrial precipitation, with higher river flow occurring during autumn compared with spring. As rivers transport terrestrial nutrients into the nearshore waters, they contribute to maintaining high primary productivity. However, it is important to note that the environmental characteristics necessary for the survival of fish species in the nearshore waters, such as temperature and salinity, are primarily determined by marine circulation rather than river runoff [22,23].

After conducting a preliminary analysis of the marine geography and location of Hainan Island's nearshore waters, the entire survey area was divided into three zones for the purpose of comparative analysis and discussion. These zones were designated as Zone I, Zone II, and Zone III. Zone I consisted of 17 stations and was adjacent to the fishing grounds of the Beibu Gulf, representing a semi-enclosed marine ecosystem. Zone II, comprising 12 stations, was located along the southern coast of the Qiongzhou Strait and Leizhou Bay. Zone III, with 21 stations, was situated in the northern part of the South China Sea. The study included 50 sampling stations distributed across these three zones (Figure 1).

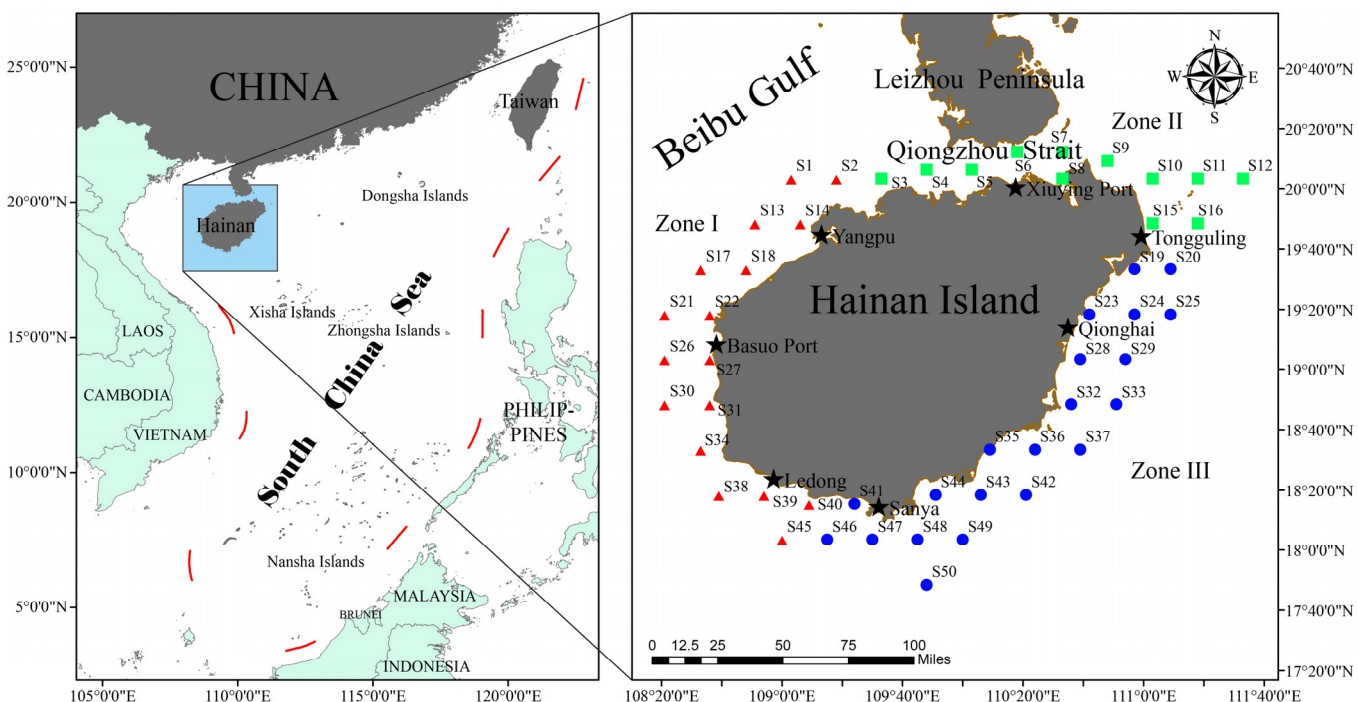

**Figure 1.** Investigation station and research division of fish resources in the Hainan coastal zone. Zone I (indicated by red triangles), Zone II (represented by green squares), and Zone III (marked as blue circles).

## 2.2. Material Sources

The data for this study were collected during two research expeditions conducted in the nearshore waters of Hainan Island in the spring season (May 2022) and autumn season (September 2022). Following the Specifications for Marine Fishery Resources Survey and the Technical Regulations for Marine Biological and Ecological Surveys, the survey area encompassed coordinates ranging from 108°21′00″–111°33′00″ E to 17°47′00″–20°12′00″ N. The surveys were carried out during daylight hours, with an average trawling speed of 3 knots and a trawling duration of 1 h. The investigation in this study was conducted aboard the bottom trawl fishing vessel Guibeiyu 69068, which had a main engine power of 436 kW. The trawling gear specifications included a mouth width of 37.7 m, a 20 cm mesh size at the mouth opening, and a 40 mm mesh size in the codend.

The collected catches were classified based on the latest ichthyological monographs [24–26] and FishBase [27], and on-site sample identification, weighing, counting, and biological measurements were performed after the catch was collected. The fish species were identified to the species level, and their body mass was measured with an accuracy of 0.1 g. Unidentified individuals were cryopreserved and subsequently transferred to the laboratory for classification and identification. Environmental parameters, such as water temperature, salinity, pH, and dissolved oxygen, were measured synchronously at each sampling station during the trawl survey using the YSI 5908 Membrane Kit 1.25 Mil PE. The water depth was determined using an onboard depth sounder.

*2.3. Data Analysis*

2.3.1. The Index of Relative Importance (*IRI*)

The ecological dominance of fish communities during different seasons was evaluated using the index of relative importance (*IRI*) [4,28].

$$IRI = (N + W) \times F \times 10,000 \tag{1}$$

The *IRI* is computed by considering three factors: *N*, which represents the percentage of individuals of a particular fish species relative to the total catch; *W*, which represents the percentage of the weight of that fish species relative to the total catch weight; and *F*, which represents the percentage of occurrences of that fish species relative to the total number of sampling stations. Species with an *IRI* value $\geq$ 1000 are categorized as dominant, species with *IRI* values between 100 and 1000 are classified as important, species with *IRI* values between 10 and 100 are considered common, and species with *IRI* values < 10 are regarded as rare.

2.3.2. Estimates of Fish Resource Density

Fish resources were estimated using the bottom trawl survey method, which incorporates biomass (kg/km$^2$) and abundance (ind/km$^2$) measurements [29].

$$U = d/(a \times q) \tag{2}$$

The estimation formula is defined as follows: *U* denotes the fish biomass or abundance, measured in kg/km$^2$ or ind/km$^2$; *d* represents the average catch per hour in kilograms or individuals per hour; *a* indicates the trawl swept area per hour in square kilometers per hour; and *q* represents the escape rate (for this study, a value of 0.5 was employed).

2.3.3. Diversity Index Calculation

The Shannon-Wiener diversity index (*H'*) was used in this study to calculate community species diversity based on biomass density [30]. The improved formula proposed by Wilhm [31] was employed for the calculation, which is defined as follows:

$$H\prime = -\sum_{i=1}^{s} (P_i \ln P_i). \tag{3}$$

Pielou evenness index (*J'*):
$$J\prime = H\prime / \ln S. \tag{4}$$

Margalef species richness index (*D*):

$$D = (S - 1)/\ln N. \tag{5}$$

In the formula, *S* represents the total number of species of captured fish in the study area, $P_i$ denotes the weight of the *i*th fish species, *N* represents the total number of individuals of captured fish, *J'* corresponds to the evenness index, and ln*S* signifies the maximum value of the diversity index.

2.3.4. ANOSIM and SIMPER Analysis

We utilized cluster analysis to investigate the community structure based on the Bray–Curtis similarity coefficient matrix. The fish abundance data were subjected to a square root transformation prior to calculation [32]. To assess the significance of variations in community structure among different groups, we applied the one-way analysis of similarity (ANOSIM) approach using fish species composition and relative abundance data [32]. Additionally, SIMPER analysis was conducted to determine the contributions of specific species to the within-group similarity and between-group dissimilarity of the community structure [32,33]. All computational analyses were performed using Primer 5.0.

2.3.5. Abundance Biomass Curve Analysis

The k-dominance curve, a useful tool for assessing environmental pollution status, was employed in this study [34]. Specifically, the Primer 5.0 software was used to generate k-dominance curves (abundance/biomass curve, ABC) depicting the abundance and biomass of fish for the two survey cruises. A comparative analysis was also conducted to examine the relationship between these curves [32]. The process of individually plotting the ABC for each sample can be laborious, especially when dealing with multiple sampling stations, time points, or replicates. To streamline the procedure and facilitate statistical analysis, Clarke [35] introduced a simplified measure known as *W*. The *W* statistic ranges from $-1$ to $+1$, where a positive value indicates a scenario in which species abundance is evenly distributed, but biomass is dominated by a single species. Conversely, a negative value of *W* signifies the opposite pattern.

2.3.6. Canonical Correspondence Analysis

Prior to conducting canonical correspondence analysis (CCA), we performed de-trended correspondence analysis (DCA) to calculate the lengths of gradient axes. If the maximum gradient value among the four axes was less than 3, we selected redundancy analysis for analysis. If it exceeded 4, CCA was chosen. If it fell between 3 and 4, both methods can be used [36]. The environmental factors considered in this study included bottom-layer salinity (BSS), surface-layer salinity (SSS), bottom-layer temperature (BST), surface-layer temperature (SST), water depth (depth), pH, dissolved oxygen (DO), and chlorophyll-a concentration (Chl-a).

Given that bottom trawling was employed for the survey, pH and DO data from the bottom layer were utilized. The chlorophyll-a data were obtained from the MODIS satellite by the National Aeronautics and Space Administration at NASA Ocean Color [37]. To ensure more accurate results and minimize interference from dominant and opportunistic species, fish species with fewer than 10 individuals in both the spring and autumn seasons were excluded. As a result, 172 species (spring) and 153 species (autumn) were retained, representing 60.99% and 54.64% of the total number of fish species in each season, respectively. Additionally, a logarithmic transformation was applied to the fish abundance and environmental factor data.

## 3. Results

### 3.1. Species Composition and Dominant Species

A total of 363 fish species were captured during this investigation, representing 24 orders, 114 families, and 226 genera (Table 1, Table S1); fish species from families such as Sciaenidae, Pleuronectidae, and Anguillidae exhibit significant economic value. However, despite their high diversity, these species are characterized by smaller size and lower numeric abundance. Osteichthyes accounted for 95.59% of the total fish species, comprising 18 orders, 105 families, 217 genera, and 347 species. The dominant order was Perciformes, with 56 families, 107 genera, and 181 species, constituting 49.86% of the overall fish species. The second most abundant order was Scorpaeniformes, consisting of 8 families, 31 genera, and 39 species (10.74%). Pleuronectiformes followed with 7 families, 19 genera, and 34 species (9.47%), and Anguilliformes with 8 families, 13 genera, and 22 species

(6.06%). Fish species from other orders were relatively scarce, comprising less than 5% of the total fish species.

**Table 1.** Fish species composition in coastal waters of Hainan Island.

| Class | Order | Family | Genus | Species |
|-------|-------|--------|-------|---------|
| Chondrichthyes | Carcharhiniformes | 3 (2.63%) | 3 (1.33%) | 3 (0.83%) |
| | Orectolobiformes | 1 (0.88%) | 1 (0.44%) | 1 (0.28%) |
| | Myliobatiformes | 1 (0.88%) | 1 (0.44%) | 4 (1.10%) |
| | Rajiformes | 2 (1.75%) | 2 (0.88%) | 5 (1.38%) |
| | Torpediniformes | 1 (0.88%) | 1 (0.44%) | 2 (0.55%) |
| Cyclostomata | Myxiniformes | 1 (0.88%) | 1 (0.44%) | 1 (0.28%) |
| Osteichthyes | Clupeiformes | 3 (2.63%) | 9 (3.98%) | 15 (4.13%) |
| | Myctophiformes | 1 (0.88%) | 3 (1.33%) | 8 (2.20%) |
| | Siluriformes | 2 (1.75%) | 3 (1.33%) | 3 (0.83%) |
| | Anguilliformes | 8 (7.02%) | 13 (5.75%) | 22 (6.06%) |
| | Mugiliformes | 2 (1.75%) | 2 (0.88%) | 5 (1.38%) |
| | Beryciformes | 1 (0.88%) | 1 (0.44%) | 1 (0.28%) |
| | Scorpaeniformes | 8 (7.02%) | 31 (13.72%) | 39 (10.74%) |
| | Perciformes | 56 (49.12%) | 107 (47.35%) | 181 (49.86%) |
| | Tetraodontiformes | 2 (1.75%) | 11 (4.87%) | 15 (4.13%) |
| | Gasterosteiformes | 3 (2.63%) | 3 (1.33%) | 4 (1.10%) |
| | Gadiformes | 3 (2.63%) | 3 (1.33%) | 4 (1.10%) |
| | Pleuronectiformes | 7 (6.14%) | 19 (8.41%) | 34 (9.37%) |
| | Lophiiforme | 4 (3.51%) | 5 (2.21%) | 9 (2.48%) |
| | Ophidiiformes | 1 (0.88%) | 3 (1.33%) | 3 (0.83%) |
| | Zeiformes | 1 (0.88%) | 1 (0.44%) | 1 (0.28%) |
| | Salmoniformes | 1 (0.88%) | 1 (0.44%) | 1 (0.28%) |
| | Gonorhynchiformes | 1 (0.88%) | 1 (0.44%) | 1 (0.28%) |
| | Aulopiformes | 1 (0.88%) | 1 (0.44%) | 1 (0.28%) |

At the family level, Carangidae exhibited the highest species richness with 23 species, followed by Serranidae with 16 species. Bothidae and Scorpaenidae ranked third and fourth, respectively, with 14 and 13 species. The remaining families had fewer than 10 species. Chondrichthyes, which includes cartilaginous fish, comprised 5 orders, 8 families, 8 genera, and 15 species, representing 4.13% of the total fish species. Myliobatiformes had the highest species richness with 5 species, while Orectolobiformes had the fewest with only one species, and Cyclostomata had 1 species.

In terms of ecological dominance, during the spring season, the dominant species ($IRI \geq 1000$) were *Acropoma japonicum* and *Decapterus maruadsi*. Additionally, there were 7 important species ($100 \leq IRI < 1000$), namely, *Navodon xanthopterus*, *Trachurus japonicus*, *Upeneus japonicus*, *Psenopsis anomala*, *Saurida tumbil*, *Champsodon atridorsalis*, and *Saurida undosquamis*. The cumulative abundance and biomass of the top 5 dominant species accounted for 51.76% and 82.92% of the total abundance and biomass, respectively. In the autumn season, there were no dominant species ($IRI \geq 1000$), but 17 important species ($100 \leq IRI < 1000$) were observed, including *Leiognathus bindus*, *S. tumbil*, *C. atridorsalis*, *Leiognathus berbis*, *Johnius belengeri*, *S. undosquamis*, *Pennahia macrocephalus*, *A. japonicum*, *U. japonicus*, *Brachypleura novaezeelandiae*, *Pennahia anea*, *D. maruadsi*, *Therapon thraps*, *Rogadius asper*, *Upeneus sulphureus*, *Ilisha melastoma*, and *Parargyrops edita*. The cumulative abundance and biomass of the top 5 important species accounted for 16.38% and 31.42% of the total abundance and biomass, respectively. *A. japonicum*, *D. maruadsi*, *U. japonicus*, *S. tumbil*, *C. atridorsalis*, and *S. undosquamis* were identified as dominant species in both the spring and autumn seasons.

### 3.2. Spatial and Temporal Variation in Fish Stocks

Through sampling and analyzing the spring catches in the coastal waters of Hainan Island, this study revealed a significant variation in fish biomass within the study area, with an average biomass of 1074.32 kg/km². Station S39 exhibited the highest biomass, while the lowest biomass was observed at station S5. Figure 2a illustrates the uneven distribution of fish biomass observed during the spring survey. The areas with high biomass are predominantly concentrated in Zone I, whereas the biomass near the Qiongzhou Strait deviates significantly from the average value.

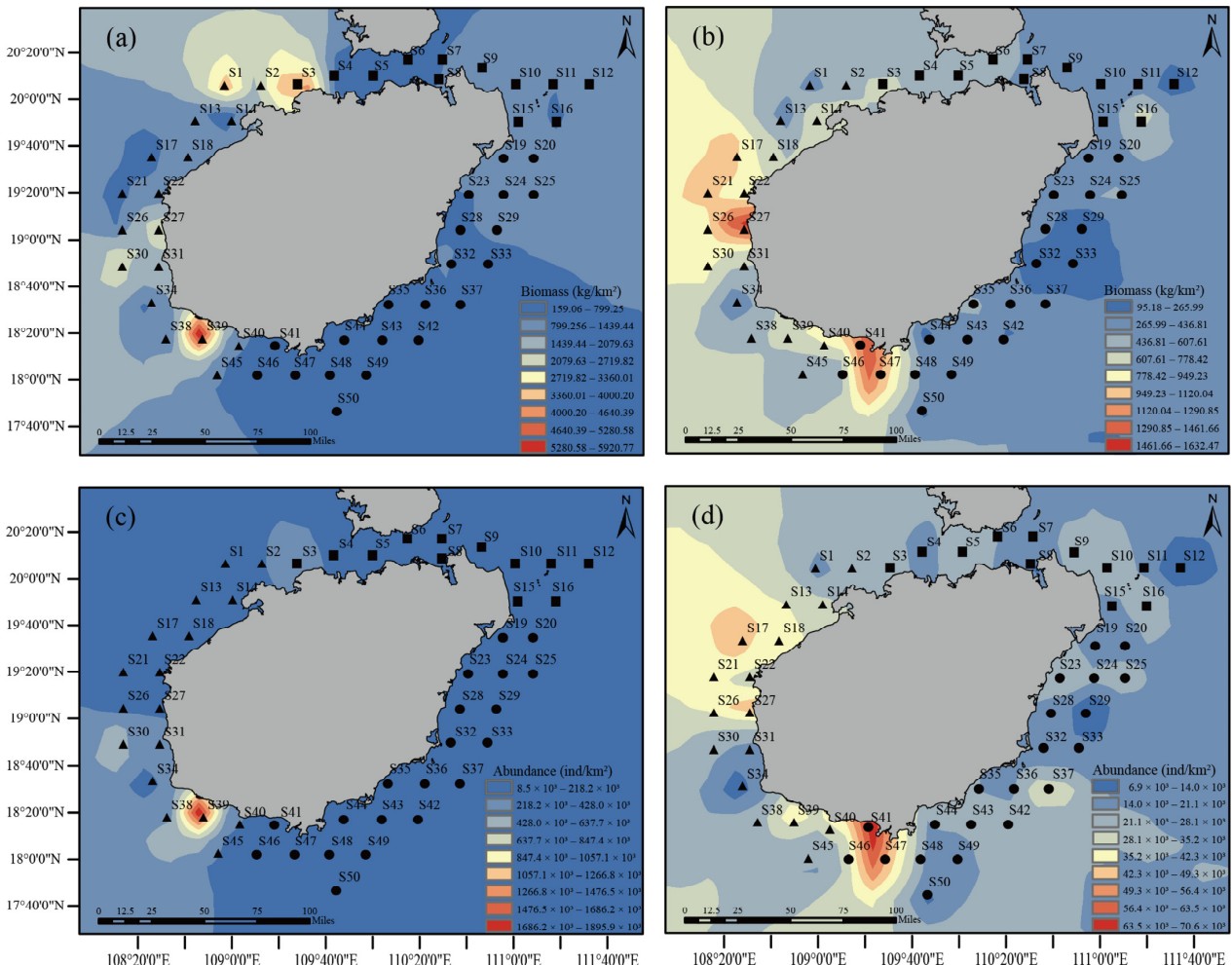

**Figure 2.** Temporal and spatial distribution of fish resources in coastal waters of Hainan Island. (**a,c**) represent the biomass and abundance of the inshore waters of Hainan Island in spring, (**b,d**) are for autumn.

Species abundance showed a similar pattern to biomass but exhibited an uneven spatial distribution, with an average of $1.24 \times 10^5$ ind/km². High-abundance areas were identified at stations S38 and S39 in Zone I, while survey stations in Zone I (S1, S2, S13, and S18) exhibited abundances ranging from 1.12 to $2.98 \times 10^5$ ind/km². Low-abundance areas were observed in the northern part of Zone II and the central part of Zone III (Figure 2c).

During autumn, the average biomass of the captured fish was 534.87 kg/km², with a range of 20.66 (S29) to 2179.92 kg/km² (S27). The spatial distribution of fish biomass showed an uneven pattern, with concentrated high levels primarily found in the southern regions of Zones I and III. In contrast, the biomass in Zone II and the central part of Zone III was significantly lower than the average, particularly at stations such as S12, S29, S32, and S33 (Figure 2b).

The distribution of species abundance also exhibited irregularity, with an average of $0.26 \times 10^5$ ind/km$^2$ and a range of 3236.52 to $0.849 \times 10^5$ ind/km$^2$. Specifically, stations S47 and S41 displayed higher abundance levels than other sites, surpassing the average abundances. In contrast, station S29 exhibited relatively poor abundance, representing the lowest values. Although over 60% of the survey stations did not reach the average abundance, relatively high-value areas were observed in the central part of Zone I (stations S17, S18, and S27) and the southern part of Zone III (stations S41 and S47) (Figure 2d).

### 3.3. Diversity Index

Figure 3 shows the diversity index variations in the study area. Generally, the Shannon-Wiener diversity index ($H'$), species richness index ($D$), and evenness index ($J'$) exhibited significant variations among different stations, demonstrating a general pattern of higher values in the southeast and lower values in the northwest.

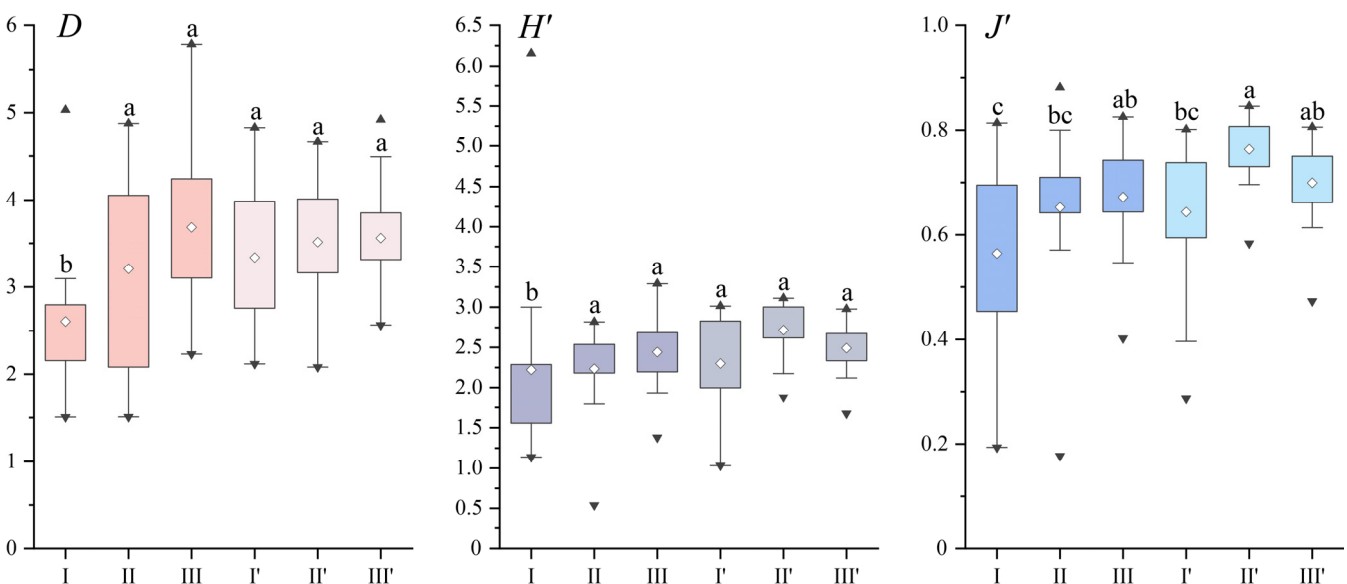

**Figure 3.** Diversity index of the fish community in the coastal waters of Hainan Island (I, II, and III for spring; I$'$, II$'$, and III$'$ for autumn). Maximum, minimum, and average values are indicated by a black triangle, black inverted triangle, and white rhombus, respectively. Different lowercase letters in the figures represent the significant differences at $p < 0.05$.

Regarding seasonal fluctuations, the Shannon-Wiener diversity index ($H'$) ranged from 0.54 to 6.16 in both spring and autumn, with average values of 2.32 and 2.48, respectively. Notably, the diversity index increased during autumn, with certain stations recording values between 2.50 and 3.10. The spatial distribution also differed, with elevated values near the Qiongzhou Strait, Yangpu, and Qionghai, while lower values were observed near Sanya, Basuo Port, and Ledong. The species richness index ($D$) ranged from 1.51 to 5.79, with higher concentrations in the northern ports of Zone III and Tongguling during spring, whereas Zones I and II exhibited lower values ranging from 1.50 to 2.50. The evenness index ($J'$) ranged from 0.18 to 0.88, primarily showing higher values in Zone III and Zone II and lower values near Basuo Port in Zone I. During autumn, the average value of the evenness index ($J'$) slightly increased, and the spatial distribution aligned generally with that of spring.

One-way analysis of variance revealed significant differences in the species richness index ($D$) and evenness index ($J'$) among the three zones during spring ($D$: $F = 4.021$, $p = 0.024$; $J'$: $F = 4.476$, $p = 0.017$) and highly significant differences in the Shannon-Wiener diversity index ($H'$) ($H'$: $F = 9.939$, $p = 0.01$). In autumn, a significant difference was found in the evenness index ($J'$) ($J'$: $F = 4.671$, $p = 0.014$), while the Shannon-Wiener diversity index ($H'$) and species richness index ($D$) showed no significant differences.

### 3.4. Community Stability

The fish communities in the coastal waters of Hainan Island have experienced varying degrees of disturbance across different survey periods and regions. Based on the ABC shown in Figure 4, the *W* values range from $-0.113$ to $-0.046$, with the lowest *W* statistic value observed in Zone II during autumn (Figure 4e), while the highest value was recorded in Zone III (Figure 4f). Specifically, during spring, the abundance curve for Zones I (Figure 4a) and II (Figure 4b) starts at a higher point, indicating the dominance of one or a few opportunistic species in the population and resulting in a significant disturbance to the community. Conversely, in autumn, the biomass curve for Zone III (Figure 4f) begins above the abundance curve and shows a clear crossing, suggesting a moderate level of disturbance to the fish community.

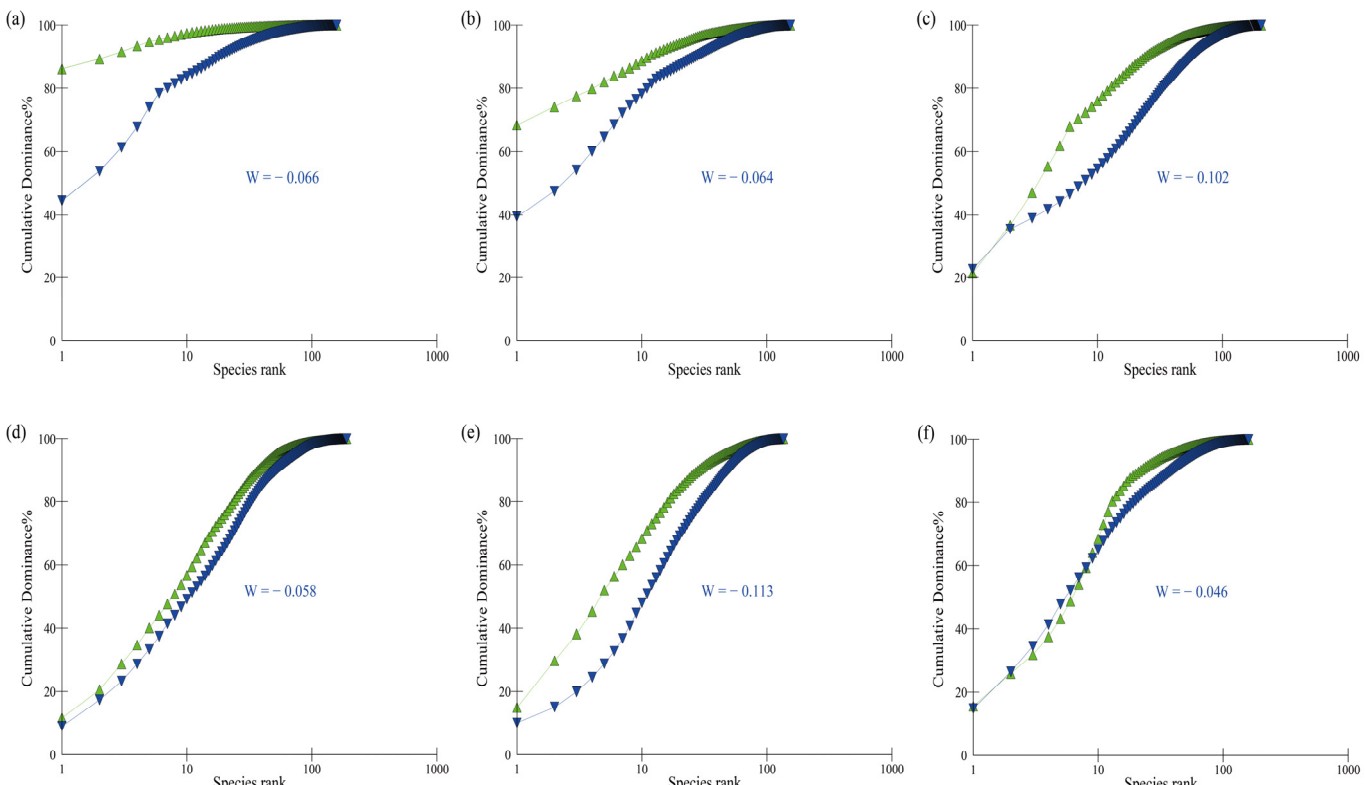

**Figure 4.** Fish abundance/biomass curve (ABC) in the coastal waters of Hainan Island. (**a–c**) for Spring Zone I, II, and III; (**d–f**) for Autumn Zone I, II, and III. Green symbols represent abundance; blue symbols represent biomass.

In summary, the fish communities in the coastal waters of Hainan Island have encountered a relatively high degree of disturbance, particularly in the eastern part of Zones II and III, which experienced more pronounced impacts compared with other areas.

### 3.5. Correlation of Communities with Environmental Factors

The correlation analysis results for seasonal environmental factors are presented in Figure 5a. During the spring season, there was a highly significant positive correlation between surface salinity (SSS) and bottom salinity (BSS) ($R = 0.89$, $p < 0.01$). Conversely, the bottom salinity exhibited the most significant negative correlation with chlorophyll-a concentration (Chl-a) ($R = -0.69$, $p < 0.01$). In autumn (Figure 5b), the surface salinity showed the most significant positive correlation with the bottom salinity ($R = 0.89$, $p < 0.01$), while the depth displayed the most pronounced negative correlation with the bottom water temperature (BST) ($R = -0.82$, $p < 0.01$).

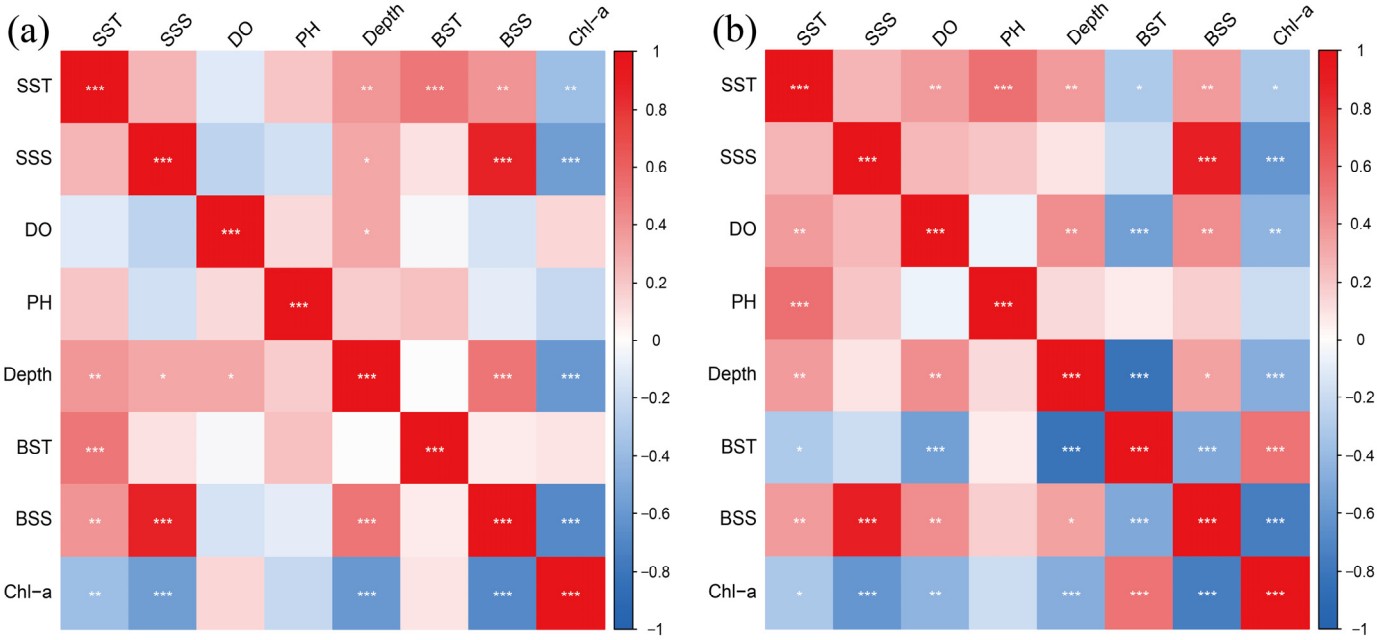

**Figure 5.** Correlation heatmap of environmental factors in the coastal waters of Hainan Island in spring (**a**) and autumn (**b**) (*** $p < 0.01$, ** or * $p < 0.05$).

The results of the DCA analysis for the spring season revealed gradient lengths of 4.51, 6.35, 3.89, and 3.37 for the four axes (Table 2), all of which exceeded a value of 4. These findings indicate a unimodal relationship in the response of fish community parameters to environmental factors in the coastal waters of Hainan Island. Consequently, CCA analysis is appropriate for examining the association between fish community composition and environmental factors in this region. The first and second axes were identified as the primary component axes in the CCA analysis (Table 3). The first axis accounted for 7.72% of the variation, and the cumulative explained variation for the first two axes was 13.22%. The species–environment correlations on these first two axes were 92.38% and 84.02%, respectively.

**Table 2.** DCA analysis results of the fish community in spring and autumn.

| Number | Spring | | | | Autumn | | | |
|---|---|---|---|---|---|---|---|---|
| | Axis 1 | Axis 2 | Axis 3 | Axis 4 | Axis 1 | Axis 2 | Axis 3 | Axis 4 |
| Eigenvalues | 0.9496 | 0.7931 | 0.6362 | 0.2663 | 0.8056 | 0.6219 | 0.4001 | 0.2344 |
| Cumulative percentage explained variation (%) | 9.44 | 17.33 | 23.65 | 26.30 | 9.05 | 16.04 | 20.53 | 23.17 |
| Gradient length | 4.51 | 6.35 | 3.89 | 3.37 | 7.06 | 4.34 | 3.84 | 2.74 |

**Table 3.** CCA analysis results of the fish community in spring and autumn.

| Number | Spring | | | | Autumn | | | |
|---|---|---|---|---|---|---|---|---|
| | Axis 1 | Axis 2 | Axis 3 | Axis 4 | Axis 1 | Axis 2 | Axis 3 | Axis 4 |
| Eigenvalues | 0.7768 | 0.5530 | 0.3346 | 0.2486 | 0.5827 | 0.3348 | 0.2923 | 0.2543 |
| Cumulative percentage explained variation (%) | 7.72 | 13.22 | 16.55 | 19.02 | 6.55 | 10.31 | 13.59 | 16.45 |
| Species–environ | 0.9238 | 0.8402 | 0.7273 | 0.6052 | 0.8913 | 0.8829 | 0.7927 | 0.7800 |
| Explained fitted variation (cumulative) | 34.81 | 59.59 | 74.58 | 85.72 | 27.61 | 43.47 | 57.32 | 69.37 |

The CCA ordination plot, representing the fish community and environmental factors during the spring season in the coastal waters of Hainan Island (Figure 6a), demonstrated that among the environmental variables influencing the structure and composition of the fish community, surface water temperature exerted the strongest influence on the dominant species, followed by pH (*SST*: *pseudo-F* = 2.0, *p* = 0.004; *pH*: *pseudo-F* = 2.5, *p* = 0.006). Specifically, pH significantly impacted species such as *Muraenesox cinereus*, *Laeops parviceps*, *Cynoglossus macrolepidotus*, *D. maruadsi*, *P. anomala*, and *Branchiostegus albus*. On the other hand, *Lagocephalus gloveri*, *Synodus hoshinonis*, *Arnoglossus tenuis*, and *Bembras japonicus* exhibited a negative correlation with surface water temperature, while wide-barred angelfish and luminous damselfish displayed a positive correlation.

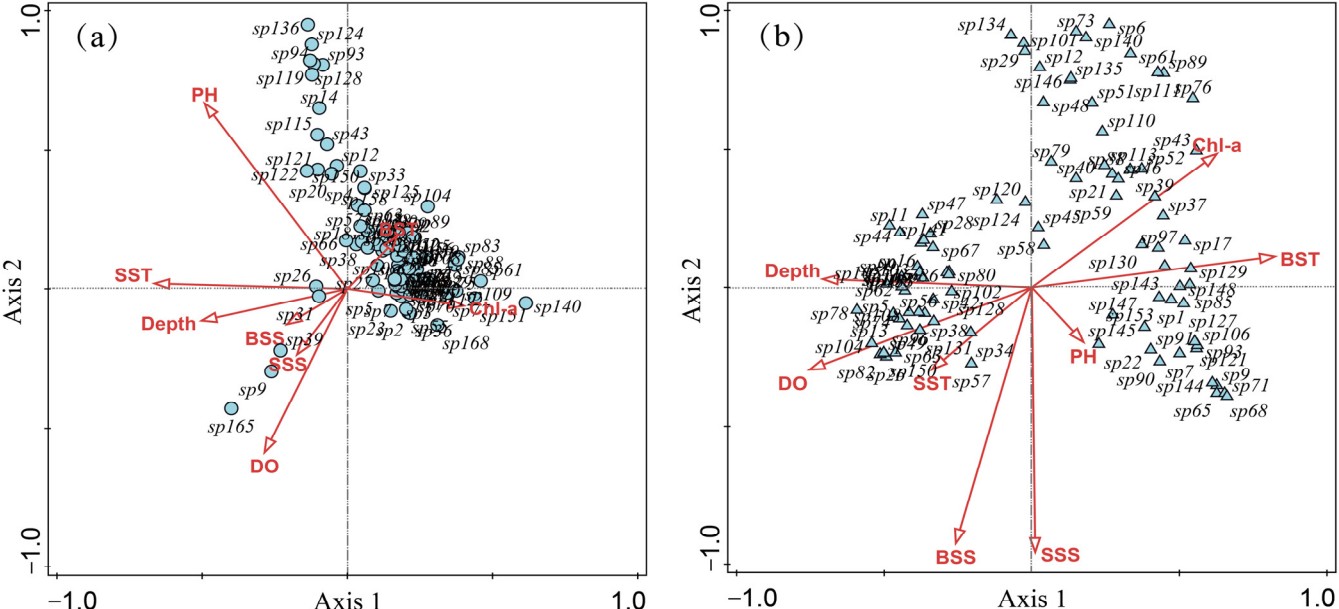

**Figure 6.** CCA analysis of the fish community and environmental factors in the coastal waters of Hainan Island in spring (**a**) and autumn (**b**).

The results of the autumn season DCA ordination demonstrated that the maximum lengths of the four ordination axes were 7.06 (Table 2). In this analysis, the cumulative explained variations for the first and second axes of species turnover were 6.55% and 10.31%, respectively. Furthermore, the correlations between species and environment with the ordination axes were 89.13% and 88.29%, respectively (Table 3). Based on these significant findings, we employed CCA analysis to further investigate the relationship between the abundance of dominant fish species and key environmental factors (Figure 6b). The aim was to identify significant environmental variables and understand their impact on the abundance of major fish species. The outcomes revealed that bottom water temperature (*BST*: *pseudo-F* = 2.7, *p* = 0.002) and bottom salinity (*BSS*: *pseudo-F* = 1.9, *p* = 0.008) were the most influential environmental factors affecting the abundance of major fish species. Specifically, a positive correlation was observed between the density of *Pomadasys maculatus*, *Apistus carinatus*, *Arius sinensis*, and *Carangoides kalla* and bottom water temperature. In contrast, *Stolephorus commersoni*, *Ostichthys japonicus*, *Halichoeres cyanopleura*, and bottom water temperature exhibited a negative correlation. *Muraenesox cinereus*, *L. bindus*, and *L. berbis* negatively correlated with bottom salinity, whereas *Pterois lunulata* and *B. novaezeelandiae* displayed a positive correlation.

## 4. Discussion

### 4.1. Spatial and Temporal Distribution of Inshore Fish Stocks on Hainan Island

Previous research has demonstrated that the cumulative impacts of ecosystem changes [38], climate change [39], and human disturbances [40,41] can result in fluctuations in the

structure of fish communities, fish diversity, and number of fish species. Our research findings corroborate this evidence. The number of recorded species of the present study (363) exceeded the 292 species documented in the 2006–2007 survey of the coastal waters of Hainan Island [42]. Moreover, it was significantly higher than the 166 species identified in the bottom trawl surveys conducted by Zhang et al. [43] during two expeditions in 2017. The current survey exhibited broader coverage and more closely spaced sampling stations compared with previous fishery resource surveys in the coastal waters of Hainan Island, resulting in a greater diversity of captured fish species.

This study reveals significant spatial variations in fish species abundance and biomass. Small-sized and low-valued species, such as *A. japonicum*, *L. bindus* flounder, and *C. atridorsalis*, exhibited a clear advantage in both seasons. These dominant species displayed distinct seasonal peaks with minimal overlap in time and space, consistent with the findings of Amara et al. [44]. The sea area surrounding S1–S3 exhibits a higher biomass during the spring season. This can be attributed to the dominance of the *D. maruadsi* and *T. japonicas* species and the nutrient-rich environment. Spring serves as the spawning and reproductive season for these species, leading to clustering behavior and consequently contributing to the higher biomass. During spring, the southwestern part of Hainan Island exhibited higher abundance and biomass, while other regions showed lower values. In autumn, apart from the peak area in the southwestern part of Hainan Island, another high-value area was observed in the western waters, particularly at stations S26 and S27. Intensified human activities aimed at accelerating economic development, including coastal tourism, agricultural production, and aquaculture, might be the major factors affecting the distribution pattern of the fish community in the southeastern coastal waters of Hainan Island [45,46]. Industrial land-based pollution resulting from development in the western waters [47], port construction in the northeastern and southern areas, and marine engineering projects [48,49] have contributed to ecological imbalance, nutrient enrichment, eutrophication, and ecological damage in the coastal marine ecosystems. Furthermore, the biomass of planktonic organisms (187.63 ind/m$^3$) might be an important contributing factor that influences the spatial distribution of fish communities [50], leading to high-value areas in the southwestern region during autumn.

Overall, the coastal waters near the western part of Hainan Island exhibited relatively high overall fish biomass, with both biomass and abundance demonstrating uneven spatial distribution during spring and autumn, highlighting significant spatial heterogeneity. Fish abundance and biomass were higher in spring, likely due to the spawning season when species such as *J. belengeri* and *Argyrosomus aneus* round herring migrate to the coastal areas of Hainan Island for spawning, replenishing the fish abundance. After the closed-season fishing, fish generally exhibited increased average weight as they had access to more abundant food resources and opportunities for self-repair and growth. Additionally, human society's growing market demand for fishery products and commercial fishing activities have significantly impacted fish resources [51], contributing to seasonal variations. These findings contribute to a deeper understanding of the ecological status and resource utilization in this region.

### 4.2. Fish Community Structure Characteristics

The average diversity index (*H'*), evenness index (*J'*), and richness index (*D*) in the coastal waters of Hainan Island were higher during autumn compared with spring, which aligns with the findings of Zhang et al. [43]. However, this survey revealed higher catches in spring. This could be attributed to the combined influence of the Kuroshio Current and the South China Sea Warm Current [18], as well as the increased temperatures and the spawning season of multiple fish species, resulting in a higher proportion of juveniles in the area.

The Fisheries Law of the People's Republic of China and the Action Plan for the Conservation of Aquatic Biological Resources of the People's Republic of China stipulate a fishing ban in the South China Sea region from May to August. Nevertheless, this

study conducted the autumn survey in mid-to-late September. With the end of the fishing moratorium, there was a significant rise in fishing pressure, selective fishing targeting major species by fishermen, and a decline in the proportion of major species, consistent with the concept of "fishing down the food web" proposed by Pauly et al. [52]. As upper-trophic-level fish species are heavily exploited, humans continue to exploit lower levels of the food web. Moreover, abnormal disturbances in the environmental climate [53] lead to a rapid decline in traditional spawning grounds and foraging areas for fish. This has led to the acknowledgment that climate change and the direct and indirect impacts of fishing can profoundly and swiftly alter the composition of marine fish communities [54]. Furthermore, under high fishing pressure, species diversity indices may exhibit an increasing trend [55], ultimately leading to a transformation of fish communities.

Analysis of similarities (ANOSIM) is a robust nonparametric hypothesis-testing framework for differences in resemblances among groups of samples [56]. The results of ANOSIM demonstrated significant variations in species composition among fish communities in the spring and autumn seasons. The overall pattern revealed a transition from long-lived, high-trophic-level, high-quality, economically valuable, and larger-sized fish species to short-lived, low-trophic-level, low-quality, economically less valuable, and smaller-sized fish species, resulting in a notable turnover of dominant species [57]. Furthermore, the SIMPER analysis highlighted that the dominant species within each season and the dissimilar species between communities were primarily the same. For instance, *A. japonicum*, *D. maruadsi*, *U. japonicus*, *L. bindus*, and *C. atridorsalis* were both representative and distinctive species within the communities. Despite the significant or highly significant outcomes obtained from the similarity analysis, these discrepancies can be attributed to spatial variances in species distribution and the inherent randomness of bottom trawl surveys [58].

*4.3. Relationships between Inshore Fish Communities and Environmental Factors on Hainan Island*

The spatial patterns of fish communities are closely associated with various environmental factors, including water temperature, depth, salinity, nutrients, and currents, as well as their own habitat preferences [59,60]. Previous studies have identified temperature and BSS as the primary environmental factors influencing the fish community structure in both the Río de la Plata estuary and the waters surrounding Zmiinyi Island [61,62]. Water temperature and salinity are widely recognized as key determinants of the spatial and temporal distribution of fish, and significant changes in these factors can gradually reshape the fish community structure [63–65].

Our CCA analysis further confirmed that surface water temperature and BSS were the principal environmental factors shaping the fish community structure during the spring and autumn seasons in the coastal waters of Hainan Island. The coastal waters of Hainan Island are primarily influenced by coastal currents and inflows from nearby rivers. The Qiongzhou Strait and its adjacent waters are characterized by low temperature and salinity, while sediment discharge from the Nandu River plays a significant role in the siltation process of the Qiongzhou Strait and Xiuying Port [19]. The discharge of sediment, in conjunction with water flow, facilitates the transportation of significant quantities of nutrients and particulate organic matter, thereby providing ample food resources for benthic and sediment-dwelling fish species. As a result, the dominant species found in Zone II communities consist of *Cynoglossus macrolepidotus* and *Gastrophysus spadiceus*. These findings align with Sutcliffe's [66,67] proposition that river discharges can indirectly impact the sustainable utilization of fishery resources by influencing crucial processes, such as fish species reproduction, growth, and survival.

From July to September, the South China Sea's western boundary current enters the southern waters of Hainan Island from the south and subsequently turns northeastward [68]. As it traverses the region, this sea area exhibits high temperature and salinity characteristics. The typical species found in the Zone III community consist of *S. tumbil*, *S. undosquamis*, and *B. novaezeelandiae*, which are warm-temperate fish species. Zone I,

primarily influenced by circulation within the Beibu Gulf [69,70], encompasses a sea area with high temperature and salinity within a defined range (Figure S1). Additionally, numerous freshwater rivers flow into this region, resulting in a more intricate hydrological environment. The dominant species composition within this region includes not only small pelagic fish species, such as *L. bindus*, but also fish species of higher trophic levels and economic value, such as *A. aneus* and *J. belengeri*. Larger-sized fish species tend to inhabit and spawn in areas with abundant food resources and optimal water temperature and salinity conditions, while smaller-sized fish species with limited swimming abilities are more susceptible to the influence of factors such as ocean currents and environmental conditions.

## 5. Conclusions

This study systematically investigated the fish resources of Hainan Island, addressing previously lacking fundamental issues, such as species composition and geographical distribution in the coastal waters. The findings revealed distinct seasonal shifts in dominant fish species within the coastal waters of Hainan Island. The fish community was characterized by a preference for small-sized individuals, a simple age structure, and low trophic levels, suggesting a declining trend of fish resources. Furthermore, diversity indices exhibited a significant increase during autumn, with temperature and salinity identified as influential environmental factors shaping the spatial distribution of fish communities. This study holds a considerable reference value for developing scientifically sound fishery management policies and promoting sustainable resource utilization. Further research should focus on specific characteristics of Hainan Island's coastal waters, except for temperature and salinity, such as comprehensive disturbances caused by coastal industrial activities, wastewater discharge, and noise pollution to fish populations.

**Supplementary Materials:** The Supplementary Material for this article can be found online at https://www.mdpi.com/article/10.3390/d15080901/s1, Table S1: List of identified fish species in the coastal waters of Hainan Island; Table S2: Dominant species (*IRI* > 100) in the nearshore waters of Hainan Island; Table S3: Hainan Island nearshore marine diversity index. (I, II, III for Spring; I′, II′, III′ for Autumn); Table S4: Differences in the composition of community structure among coastal waters of Hainan Island in spring; Table S5: Differences in the composition of community structure among coastal waters of Hainan Island in autumn; Table S6: Typical species (contribution rate > 4%) and their contribution to the mean similarity within the group within each sea area in spring on Hainan Island; Table S7: Divergent species (contribution rate > 4%) and their contribution to intergroup mean dissimilarity among different seas in spring on Hainan Island; Table S8: Typical species (contribution rate > 4%) and their contribution to the mean similarity within the group within each sea area in autumn on Hainan Island; Table S9: Divergent species (contribution rate > 4%) and their contribution to intergroup mean dissimilarity among different seas in autumn on Hainan Island; Figure S1: Spatial and temporal variability of biotic (abundance, biomass) and abiotic (temperature, salinity) factors in the nearshore waters of Hainan Island. Marine Distribution: I, Zone I; II, Zone II; III, Zone III. (a,c,e,g) for spring; (b,d,f,h) for autumn.

**Author Contributions:** Conceptualization, Z.L. and T.G.; original draft, Z.L.; data curation, L.W.; funding acquisition, T.G. and D.S.; validation, Y.L. and M.L.; visualization, C.C. and B.S.; writing—review and editing, C.Y. and D.S. All authors have read and agreed to the published version of the manuscript.

**Funding:** This study was supported by the Hainan Provincial Natural Science Foundation of China (Grant No. 320QN358) and the Hainan Provincial Government Purchases Services Project (Grant No. HNDW2020-112).

**Institutional Review Board Statement:** Not applicable.

**Data Availability Statement:** The original contributions presented in the study are included in the article/Supplementary Material. Further inquiries can be directed to the corresponding authors.

**Conflicts of Interest:** The authors declare no conflict of interest.

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
