# Peer review of "Relationships between Fish Community Structure and Environmental Factors in the Nearshore Waters of Hainan Island, South China"

_diversity, doi:10.3390/d15080901_

Round 1
Reviewer 1 Report (Previous Reviewer 1)
I noticed authors followed my suggestions. However, I have some recommendations:
Line 69, "The objective of this study aim to..." It must simply be: The objectives of this study were... or This work aimed to...
The commercial value of fish MUST be explained in the Materials and methods section instead of the Results section. So, move text from lines 207-209 to the Mat and meth section, please.
Moderate revision, please.
Author Response
Responses to reviewer 1
- Line 69, "The objective of this study aim to..." It must simply be: The objectives of this study were... or This work aimed to...
Response: Thank you for your kind comments. We have amended line 69.
- The commercial value of fish MUST be explained in the Materials and methods section instead of the Results section. So, move text from lines 207-209 to the Mat and meth section, please.
Response: We sincerely appreciate your valuable feedback on the paper. We acknowledge the significance of this point. However, after careful consideration of the data analysis, we maintain that lines 207-209 should remain in the “Results” section. Our reasons for this choice are as follows:
Firstly, the section on data analysis and results serves as a dedicated segment to present the pivotal findings and results of the study. Since the statement of the business value is rooted in the results and findings of the data analyses, its placement in the Results section effectively communicates the importance and significance of these findings to the reader.
Secondly, including the commercial value in the Materials and Methods section may cause confusion, as this section typically focuses on providing a detailed account of the experimental design, methodology, and data collection, rather than facilitating a discussion of the results and conclusions.
Based on these reasons, we recommend that lines 207-209 remain in their original position, namely the “Results” section. By doing so, we are confident that the presentation of the study’s results will be enhanced and will align with the reader’s expectations. Once again, we express our gratitude for your thorough review and insightful comments on our paper.

Reviewer 2 Report (Previous Reviewer 2)
General comments
The authors collect the fish community structure around Hainan Island and evaluate the relationships between fishes and environmental factors. This study will be an important knowledge for the management of fish diversity and resources in the study area. However, there are some uncertainties in the manuscript. In particular, the inadequate discussion and explanation in the “Discussion part” and “Figure Legends” need to be improved. The authors would need to revise their manuscript to eliminate these ambiguities and make it more understandable. Thus, the substantial revision is needed to make this manuscript suitable for publication.
Specific comments
L39-40 This expansion has led to~
Fish diversity and fish resources should include non-benthic taxa such as swimming fish, but why focus on benthic fish in this study?
L56-64 Located in the subtropical and tropical regions,
"Understanding the potential impact of human activities" in this study, so please provide some background on human activities on Hainan Island and surrounding coastal waters.
L99-106 After conducting a preliminary analysis~
What criteria were used to distinguish these three types of areas?
For example, is it environmental similarity?
Or is it a fisheries management area?
In the manuscript and in Figures, the names of the areas are expressed with various words, such as "Zone," "area," "Region".
L113
The survey area~ => the survey area
L121
Search Fish Base => FishBase
Also, please cite FishBase using the correct citation as stated on their website (https://fishbase.mnhn.fr/search.php).
L192
The abbreviation should also be used for surface-layer salinity (SSS?), since it is also used in the figure.
L225-239 In terms of ecological dominance, ~
Where is the information listed here available?
L248-249 The areas with high biomass are~
All areas with high biomass in spring are in Zone I (around S1-3 and S39), where do the authors explain the results of high biomass in Zone III?
L249 Qiongzhou Strait
What area is this on the map?
Since it is not listed in either Figure 1 or Figure 2, I recommend that you use the name of a nearby point or some other method to describe it.
L282-293
There should be a Table to which information such as diversity indices can be referred. At least, it should be supplied as Supplementary material.
In addition, proper nouns (Qiongzhou Strait, Yangpu, etc.) are not shown on the map. Therefore, it is necessary to use the name of the site name (or proper noun + site name) instead of the proper noun to explain the location.
L322 surface salinity and bottom salinity~
It would be better for the reader to add the abbreviations defined by the authors.
L324-327 In autumn (Figure 5b),~
“the most pronounced negative correlation with the bottom salinity”
Could this result be bottom water temperature (BST) rather than bottom salinity (BSS)?
In addition, since significant relationships were found between many environmental factors in spring and fall, those results should also be mentioned.
L334-335 The results of the DCA analysis for the spring season~
This representation is incorrect because not all gradient length exceeds 4.
L354-355 The results of the autumn season~
In autumn, gradient length less than 3 have also been observed.
According to the criteria used by the authors, if a value of less than 3 is included, then the CCA cannot be used.
It is very doubtful why the authors are ignoring the criteria they set themselves.
L393-394 During spring, the southwestern part of Hainan Island~
A discussion of the factors contributing to the high biomass around S1-3 would be important.
L402-407
It would be necessary to discuss how the development of industry has affected the resultant increase in fish biomass and abundance.
Industry itself does not seem to be a factor that gives rise to inter-seasonal variation, so why is this effect present only in autumn?
L412-423
Trends in each season have been discussed, but little has been discussed regarding inter-seasonal variability.
For example, spring fish resources show a regional distribution, while that of autumn shows high fish biomass and abundance over a wide area.
Why is this variation occurring?
L435-440 With the end of ~
Which data or chart explains this discussion?
L447-461 Analysis of similarities (ANOSIM) is~
Methodological explanations for ANOSIM and SIMPER are missing, please add them to Methods.
Also, please show the results of ANOSIM and SIMPER in the paper or in Supplementary Information.
L481
Region II?
L481-484 Consequently, the dominant species in~
What trends did the results of this study show and "how" were they consistent with previous studies?
Additional explanation and discussion in the Discussion part should be needed.
L485-499
This section mostly explained the facts of each coastal area, and I could not understand what the authors discuss from their results.
Figure 1
There is a lack of explanation regarding the diagram.
For example, 1) what does the red dashed line on the left indicate, 2) In the diagram on the right, which symbols (+ which colors) indicate the respective areas?
Figure 3
In the figure, the line of the lowest value should be plotted as well as the line of the highest value.
Figure 4
A legend should be included to show what the green and blue lines (symbols) in the figure represent, respectively.
Figure 5
I think that the use of pie charts in correlation makes it difficult for the reader to understand.
The correlation is well understood by the color expression (only heat map).
Author Response
On behalf of my co-authors, we thank you very much for giving us an opportunity to revise our manuscript, we appreciate editor and reviewers for their positive and constructive comments and suggestions on our manuscript. The responses to the comments as follows:
Responses to reviewer 2
- L39-40 This expansion has led to~
Fish diversity and fish resources should include non-benthic taxa such as swimming fish, but why focus on benthic fish in this study?
Response: We sincerely appreciate your thorough review and invaluable comments on our paper. The research data in our study were collected through bottom trawl surveys, encompassing three major taxa: fish, crustaceans, and cephalopods. For the purpose of this paper, we specifically analyzed the fish data.
Benthic fish constituted the majority of the captured fish, simultaneously playing a vital role in marine ecosystems and exerting significant influence on ecosystem function and structure. Due to their higher economic value compared to pelagic fish, comprehending the ecological functions and contributions of benthic fish to the ecosystem is of utmost importance. Our aim is to gain a deeper understanding of the interactions and regulatory mechanisms among benthic fish, as influenced by environmental factors, to enhance our comprehension of their ecological functionalities and contributions.
It is important to note that non-benthic taxa, such as swimmers, also hold value in assessing fish diversity and resources. Specialized survey methodologies are employed for pelagic fishes. Consequently, in future studies, we aspire to delve deeper into comprehending the ecological traits and ecosystem implications of non-benthic fishes. We will explicitly address the limitations of our current study in the Discussion section, while indicating potential research directions encompassing investigations of non-benthic fishes.
- L56-64 Located in the subtropical and tropical regions, "Understanding the potential impact of human activities" in this study, so please provide some background on human activities on Hainan Island and surrounding coastal waters.
Response: We sincerely appreciate your review and invaluable comments on our paper. Human activities in the vicinity of Hainan Island and its coastal waters have exerted substantial influences on the marine ecosystem. Our paper will extensively elucidate the potential impacts of these activities on both Hainan Island and the adjacent coastal waters. The analysis will be focused on evaluating how these impacts may influence the attainment of our study objectives. In the Introduction section, we will ensure adequate incorporation of essential background information and thoroughly contemplate the potential implications of these impacts on the interpretation of our findings. We will diligently address the points you raised during our revision process within the specified range (L55 - L74).
- L99-106 After conducting a preliminary analysis~
What criteria were used to distinguish these three types of areas?
For example, is it environmental similarity?
Or is it a fisheries management area?
In the manuscript and in Figures, the names of the areas are expressed with various words, such as "Zone," "area," "Region".
Response: We express our sincere gratitude for your review and invaluable comments on our thesis. We are thankful for your guidance, and as a result, we will incorporate the following additions into our revised thesis:
Following preliminary analyses, we have duly accounted for various factors. Firstly, environmental factors differ across different zones, encompassing temperature, water quality, hydrological conditions, and more. Additionally, there exist notable discrepancies in seafloor topography among the zones. Specifically, Zone I exhibits a relatively flat seafloor compared to the other two zones, with the east and west mouths of the Qiongzhou Strait (Zone II) being shallow and featuring deep troughs in the middle portion. Furthermore, the southeastern part of Zone III is characterized by gullies. Zone I, comprising 17 stations, directly connects to the fishing grounds of the Beibu Gulf, constituting a semi-enclosed marine ecosystem. Zone II, consisting of 12 stations, represents a fishing ground situated along the southern coast of the Qiongzhou Strait and Leizhou Bay. Lastly, Zone III encompasses 21 stations and is located in the northern part of the South China Sea.
Fig.1 The sketch of Qiongzhou Strait
Fig. 2 Geomorphological map of the seabed in the northern part of the South China Sea
Fig. 3 Submarine relief of the eastern continental shelf of the Hainan Island.
Uniform use of Zone I, Zone II, and Zone III to denote study zones. With these additional notes, we hope to be able to more clearly articulate our reasons for choosing the district classification criteria. We appreciate your guidance and suggestions and look forward to presenting you with an improved paper after revision.
Fig. 1: Zhao, H.T.; Wang, L.R.; Yuan, J.Y. Origin and time of Qiongzhou Strait. Marine Geology & Quaternary Geology. 2007, 27, 33-40. DOI:10.16562/j.cnki.0256-1492.2007.02.005.
Fig. 2: Feng, W.K.; Bao, C.W.; Chen, J.R. Preliminary study on submarine relief of the northern South China Sea. Haiyang Xuebao. 1982, 4, 462-472.
Fig. 3: Lin, M.H. Submarine geomorphology of the eastern continental shelf of the Hainan Island. Marine Geology & Quaternary Geology. 1995, 12, 37-46. DOI:10.16562/j.cnki.0256-1492.1995.04.004.
- L113 The survey area~ => the survey area
Response: Thank you for your kind comments. We have revised it.
- L121 Search Fish Base => FishBase
Also, please cite FishBase using the correct citation as stated on their website (https://fishbase.mnhn.fr/search.php).
Response: Thanks for the useful suggestions. We have revised the citations in the article and references.
- L192 The abbreviation should also be used for surface-layer salinity (SSS?), since it is also used in the figure.
Response: Thank you for your kind comments. We have revised it.
- L225-239 In terms of ecological dominance, ~
Where is the information listed here available?
Response: Thank you very much for your advice. We will add the corresponding form to the supplementary material.
Table S2 Dominant species (IRI > 100) in the nearshore waters of Hainan Island.
Species |
Spring |
Autumn |
|
IRI |
IRI |
||
Acropoma japonicum |
3013.82 |
332.49 |
|
Decapterus maruadsi |
1475.74 |
200.44 |
|
Navodon xanthopterus |
548.26 |
|
|
Trachurus japonicus |
386.92 |
|
|
Upeneus japonicus |
385.01 |
331.61 |
|
Psenopsis anomala |
321.09 |
|
|
Saurida tumbil |
217.72 |
646.08 |
|
Champsodon atridorsalis |
194.26 |
528.74 |
|
Saurida undosquamis |
147.97 |
380.45 |
|
Leiognathus bindus |
|
730.84 |
|
Leiognathus berbis |
|
451.21 |
|
Johnius belengeri |
|
392.12 |
|
Pennahia macrocephalus |
|
372.31 |
|
Brachypleura novaezeelandiae |
|
256.09 |
|
Pennahia anea |
|
240.42 |
|
Therapon thraps |
|
178.53 |
|
Rogadius asper |
|
145.78 |
|
Upeneus sulphureus |
|
118.03 |
|
Ilisha melastoma |
|
113.01 |
|
Parargyrops edita |
|
108.69 |
|
- L248-249 The areas with high biomass are~
All areas with high biomass in spring are in Zone I (around S1-3 and S39), where do the authors explain the results of high biomass in Zone III?
Response: We deeply appreciate your thorough review and valuable comments provided for our paper. We apologize for the writing error in our paper, where we mistakenly stated that the areas of high spring biomass were in zones I and III. We are grateful to you for bringing this to our attention, and we will rectify the discussion concerning the regions of elevated spring biomass to accurately depict our findings. In response, we will eliminate the mention of biomass concentration in zone III and instead provide a precise and accurate description of our findings. (L256 – L257).
- L249 Qiongzhou Strait
What area is this on the map?
Since it is not listed in either Figure 1 or Figure 2, I recommend that you use the name of a nearby point or some other method to describe it.
Response: We sincerely appreciate your comprehensive review and invaluable comments on our paper. We are grateful for bringing this issue to our attention, and as a result, we will incorporate the following changes and include additional notes in our revised paper:
We will ensure clear and explicit illustration of the areas depicted in the maps within the thesis. Since the areas are not extensively labeled in the thesis maps, we will revise and include distinct labels on Figure 1 to enhance the geo-locational information provided. This will assist readers in gaining a better comprehension of the geographical context underlying our study. Through these revisions and supplementary notes, we aim to enhance the clarity and precision of our paper’s geospatial descriptions.
We deeply appreciate your guidance and suggestions, and we eagerly anticipate sharing an enhanced version of our thesis with you following the revisions.
Figure 1 Investigation station and research division of fish resources in the Hainan coastal zone. Zone I (indicated by red triangles), Zone II (represented by green squares), Zone III (marked as blue circles).
- L282-293 There should be a Table to which information such as diversity indices can be referred. At least, it should be supplied as Supplementary material.
In addition, proper nouns (Qiongzhou Strait, Yangpu, etc.) are not shown on the map. Therefore, it is necessary to use the name of the site name (or proper noun + site name) instead of the proper noun to explain the location.
Response: We sincerely appreciate your meticulous review and invaluable suggestions regarding our thesis. Taking into account your suggestions, we will implement the following changes and provide additional explanations in the revised paper:
To complement paragraphs L282-293, we will incorporate a table containing pertinent information, including diversity indices (L305 – L306). Ensuring proper placement within the paper, the table will allow readers to delve into specific details such as the diversity index.
In accordance with your suggestion, we have revised Figure 1 to enhance the accuracy of the geographic location. This modification will enable readers to better comprehend and locate the specific geographic locations discussed in our paper. Through these revisions and additional descriptions, we endeavor to meet your request for diversity index information and enhance the clarity of geographic location descriptions on the map.
Once again, we extend our gratitude for your invaluable suggestions and guidance throughout our paper. We eagerly anticipate submitting the enhanced version of our paper to you following the revisions.
Table S3 Hainan Island nearshore marine diversity index. (I, II, III for Spring; I’, II’, III’ for Autumn).
Zone |
D |
H′ |
J′ |
||||||
MAX |
MIN |
AVG |
MAX |
MIN |
AVG |
MAX |
MIN |
AVG |
|
I |
5.03 |
1.51 |
2.61 |
6.16 |
1.14 |
2.22 |
0.81 |
0.19 |
0.56 |
I’ |
4.83 |
2.12 |
3.34 |
3.01 |
1.03 |
2.30 |
0.80 |
0.29 |
0.64 |
II |
4.88 |
1.51 |
3.21 |
2.82 |
0.54 |
2.23 |
0.88 |
0.18 |
0.65 |
II’ |
4.66 |
2.08 |
3.51 |
3.11 |
1.88 |
2.72 |
0.85 |
0.58 |
0.76 |
III |
5.79 |
2.23 |
3.68 |
3.29 |
1.38 |
2.44 |
0.83 |
0.40 |
0.67 |
III’ |
4.92 |
2.57 |
3.56 |
2.98 |
1.68 |
2.49 |
0.81 |
0.47 |
0.70 |
- L322 surface salinity and bottom salinity~
It would be better for the reader to add the abbreviations defined by the authors.
Response: Thank you for your kind comments. We have revised the article.
- L324-327 In autumn (Figure 5b),~
“the most pronounced negative correlation with the bottom salinity”
Could this result be bottom water temperature (BST) rather than bottom salinity (BSS)?
In addition, since significant relationships were found between many environmental factors in spring and fall, those results should also be mentioned. "
Response: Thank you for expressing your interest in our research and providing valuable feedback. We express our sincere appreciation for bringing this error to our attention. We acknowledge that your observation is correct and that a clerical error does exist in the paper. We will rectify this error in the revised manuscript.
As per your suggestion, we have included a comprehensive description of the apparent relationship between environmental factors in lines 329 to 336.
Thank you once again for reviewing the manuscript and for providing valuable comments. We will rectify this error in the revised manuscript to ensure the paper’s accuracy and credibility.
- L334-335 The results of the DCA analysis for the spring season~
This representation is incorrect because not all gradient length exceeds 4.
Response: Thank you for your kind comments. In the selection process between RDA analysis and CCA analysis, DCA analysis is initially conducted to yield four gradient lengths, from which the maximum value is determined. If the maximum value exceeds 4, CCA analysis is chosen; if it is below 3, RDA analysis is chosen; and if it falls within the range of 3 to 4, both analyses are considered applicable. Detailed information regarding this matter is provided in the article (L202 – L206). Related reference is listed as follows: Lepš, J.; Šmilauer, P. Multivariate analysis of ecological data using CANOCO. Cambridge university press: Cabridge, UK, 2003; pp.1-253, ISBN 0-521-81409-X.
- L354-355 The results of the autumn season~
In autumn, gradient length less than 3 have also been observed.
According to the criteria used by the authors, if a value of less than 3 is included, then the CCA cannot be used.
It is very doubtful why the authors are ignoring the criteria they set themselves
Response: Thank you for your kind comments. The response is as same as the above one.
- L393-394 During spring, the southwestern part of Hainan Island~
A discussion of the factors contributing to the high biomass around S1-3 would be important.
Response: We strongly agree with your perspective. In the revised paper, we will incorporate a comprehensive discussion to unveil the various factors that contribute to the high biomass near S1 - S3. Your suggestions will be taken into full consideration during the revision process, resulting in a more extensive and detailed discussion within the paper (L399 – L403).
- L402-407
It would be necessary to discuss how the development of industry has affected the resultant increase in fish biomass and abundance.
Industry itself does not seem to be a factor that gives rise to inter-seasonal variation, so why is this effect present only in autumn?
Response: We have incorporated your feedback and made revisions to this section (L403 –L417). However, due to the lack of specific information regarding the layout of industrial structures in the neighborhood, we are unable to provide a more in-depth analysis of the impacts discussed in this section.
- L412-423
Trends in each season have been discussed, but little has been discussed regarding inter-seasonal variability.
For example, spring fish resources show a regional distribution, while that of autumn shows high fish biomass and abundance over a wide area.
Why is this variation occurring?
Response: We sincerely appreciate your interest in and valuable comments on our study. We acknowledge your concerns, particularly regarding the discussion of inter-seasonal variation. The relationship between environmental factors and fish communities is of significant importance in our research, with a primary emphasis in section 4.3 Relationships between Inshore Fish Communities and Environmental Factors on Hainan Island. For the revised version, we will delve deeper into investigating the specific effects of environmental factors on seasonal variation.
We highly appreciate your suggestions, as they will enhance the quality and depth of our study. We will carefully consider them and provide a comprehensive response.
- L435-440 With the end of ~
Which data or chart explains this discussion?
Response: The conclusion of the fishing moratorium marks a period of intensified fishing activity and increased fishing production by vessels. Before the moratorium, there was a high abundance of juvenile fish in the sea, coupled with low economic returns for fishermen engaged in fishing production at sea. Consequently, many fishermen opted to remain in harbours and wait for the coming of the moratorium.
- L447-461 Analysis of similarities (ANOSIM) is~
Methodological explanations for ANOSIM and SIMPER are missing, please add them to Methods.
Also, please show the results of ANOSIM and SIMPER in the paper or in Supplementary Information.
Response: An explanation of the ANOSIM and SIMPER methods has been included in section 2.3.5 of the article (L180 – L188). The relevant results tables will be appended to the Supplementary Information.
Table S4 Differences in the composition of community structure among coastal waters of Hainan Island in spring.
Zone |
R |
P |
Zone I & II |
0.160 |
< 0.05 |
Zone I & III |
0.522 |
< 0.01 |
Zone II & III |
0.313 |
< 0.01 |
Table S5 Differences in the composition of community structure among coastal waters of Hainan Island in autumn.
Zone |
R |
P |
Zone I & II |
0.193 |
< 0.01 |
Zone I & III |
0.731 |
< 0.01 |
Zone II & III |
0.429 |
< 0.01 |
Table S6 Typical species (contribution rate > 4%) and their contribution to the mean similarity within the group within each sea area in spring on Hainan Island.
Species |
Zone I |
Zone II |
Zone III |
Acropoma japonicum |
42.30 |
|
|
Decapterus maruadsi |
5.24 |
4.26 |
|
Brachypleura novaezeelandiae |
5.08 |
|
|
Champsodon atridorsalis |
4.82 |
|
17.22 |
Saurida tumbil |
|
13.04 |
|
Psenopsis anomala |
|
8.35 |
|
Saurida undosquamis |
|
8.33 |
|
Apogon kiensis |
|
5.25 |
4.88 |
Paramonacanthus nipponensis |
|
5.13 |
|
Navodon xanthopterus |
|
4.03 |
15.57 |
Upeneus japonicus |
|
|
10.45 |
Table S7 Divergent species (contribution rate > 4%) and their contribution to intergroup mean dissimilarity among different seas in spring on Hainan Island.
Species |
Zone I & II |
Zone I & III |
Zone II & III |
Acropoma japonicum |
8.48 |
8.24 |
|
Decapterus maruadsi |
4.30 |
|
|
Navodon xanthopterus |
|
5.82 |
5.39 |
Upeneus japonicus |
|
4.12 |
4.07 |
Table S8 Typical species (contribution rate > 4%) and their contribution to the mean similarity within the group within each sea area in autumn on Hainan Island.
Species |
Zone I |
Zone II |
Zone III |
Pennahia anea |
11.27 |
|
|
Johnius belengeri |
8.58 |
|
|
Pennahia macrocephalus |
8.57 |
|
|
Leiognathus bindus |
8.08 |
7.50 |
6.53 |
Therapon thraps |
7.73 |
|
|
Leiognathus berbis |
|
8.49 |
|
Saurida tumbil |
|
6.59 |
13.64 |
Champsodon atridorsalis |
|
5.69 |
18.63 |
Upeneus sulphureus |
|
4.52 |
|
Decapterus maruadsi |
|
4.07 |
|
Saurida undosquamis |
|
|
11.78 |
Brachypleura novaezeelandiae |
|
|
8.29 |
Table S9 Divergent species (contribution rate > 4%) and their contribution to intergroup mean dissimilarity among different seas in autumn on Hainan Island.
Species |
Zone I & II |
Zone I & III |
Zone II & III |
Leiognathus bindus |
4.35 |
|
|
Johnius belengeri |
4.03 |
|
|
Champsodon atridorsalis |
|
5.48 |
4.89 |
Leiognathus berbis |
|
|
4.14 |
- L481 Region II?
Response: Thank you for your kind comments. We have revised the article.
- L481-484 Consequently, the dominant species in~
What trends did the results of this study show and "how" were they consistent with previous studies?
Additional explanation and discussion in the Discussion part should be needed.
Response: Thank you for your kind comments. According to Sutcliffe’s study, it was observed that river discharges could have a substantial impact on the species captured in the Gulf of St. Lawrence. The discharge of rivers can indirectly impact the sustainable utilization of fisheries resources by influencing vital processes including fish species reproduction, growth, and survival. Further research, as suggested by him, can delve into the intricate mechanisms through which river runoff affects diverse fish species, encompassing the consequences on larval survival and growth, as well as fish migration and habitat alterations. These findings will aid in the development of efficient measures for fisheries management and conservation, aimed at preserving the ecological equilibrium and promoting sustainable development of Gulf of St. Lawrence’s fisheries.
Our study reveals that river discharge supplies ample nutrients and food resources to benthic fish, thereby promoting their growth and reproduction. This finding demonstrates a connection between river discharge and nearshore fisheries, particularly estuarine fisheries, aligning with previous study outcomes.
Based on your suggestions, we provide additional clarification and discussion in the "Discussion" section (L482 – L490).
- L485-499
This section mostly explained the facts of each coastal area, and I could not understand what the authors discuss from their results.
Response: We greatly appreciate your interest in our research and the valuable feedback you have provided. We apologize for any confusion regarding the content of our discussion. In our study, alongside considering environmental factors, we also examined the distribution pattern of fish stocks from the perspective of ocean currents, proposing hypotheses and drawing inferences. We acknowledge that we may not have adequately explained this aspect within the article. It is, however, crucial for future investigations to delve more deeply into the structure of fish communities by considering large-scale ocean currents. We believe that these efforts will address your concerns and enhance the quality of our research.
- Figure 1
There is a lack of explanation regarding the diagram.
For example, 1) what does the red dashed line on the left indicate, 2) In the diagram on the right, which symbols (+ which colors) indicate the respective areas?
Response: We sincerely appreciate your suggestion. Figure 1 has been updated to include the pertinent geographic locations discussed in the text. The left side of the figure depicts the “nine-dash line” as a red dotted line, while the right side showcases the distribution of survey stations: 17 stations in Zone I (indicated by red triangles), 12 stations in Zone II (represented by green squares), and 21 stations in Zone III (marked as blue circles). Consequently, the figure depicts a total of 50 survey stations.
- Figure 3
In the figure, the line of the lowest value should be plotted as well as the line of the highest value.
Response: Based on your suggestions, we have modified Figure 3.
Figure 3 Hainan Island nearshore marine diversity index. (I, II, III for Spring; I’, II’, III’ for Autumn). Maximum, minimum and average values were indicated by black triangle, black inverted triangle and white rhombus, respectively.
- Figure 4
A legend should be included to show what the green and blue lines (symbols) in the figure represent, respectively.
Response: Based on your suggestions, we have modified Figure 4.
Figure 4 Fish abundance/biomass curve (ABC) in coastal waters of Hainan Island. (a), (b), (c) for Spring Zone I, II, III; (d), (e), (f) for Spring Zone I, II, III. Green symbols represent abundance; blue symbols represent biomass.
- Figure 5
I think that the use of pie charts in correlation makes it difficult for the reader to understand.
The correlation is well understood by the color expression (only heat map).
Response: Based on your suggestions, we have modified Figure 5.
Figure 5 Correlation heatmap of environmental factors in the coastal waters of Hainan Island in spring (a) and autumn (b). (*** P < 0.01, ** or * P < 0.05)

Round 2
Reviewer 2 Report (Previous Reviewer 2)
I carefully checked the responses and revisions, and satisfied them for the publication.
Author Response
Thanks for the comments and suggestions on our manuscript.
This manuscript is a resubmission of an earlier submission. The following is a list of the peer review reports and author responses from that submission.
Round 1
Reviewer 1 Report
*I urge authors to clearly establish the objective of the work. The aim of this work was to...........
*The fact that a lack on information of fish diversity from Hainan it is no justification per se.
*The very first sentence of the Introduction section is awkward. I urge authors to better choose another sentences to begin with. I strongly recommend authors to begin with what is known on fish diversity in the geographic region in which Hainan is located and what is know of fish in the region beyon Hainan. I strongly recommend to remove what is written in relation to the Gulf of Mexico because it is out of context. For instance, the paragraphs at the line 68 is better suited to be the beginning of the Introduction, and then include something else to justify the study. The paragraph describing Hainan is better suited to be relocated in the Materials and methods section.
*The beginning of the Materials and methods section must be like: Hainan island is located in.....(coordinate), within the biogeographic region, with temperature, salinity, etc. (quotes). Remove the heading "Material sources". Then open up a subsection called Survey design and explain how the surveys were carried out.
*Which taxonomic keys were used to identify fish species? Please quote.
*How the body mass was measured?
*The Index of Relative Importance is commonly used in stomach content analyses (Pinkas et al. 1971) Why did the authors decide to use it to describe fish species in the collections? Explain.
*Estimates of Fishery Resource Density is NOT the same as estimates of fish resources. Please, explain.
*Why authors did not include a calculation of simple relative abundance also?
*If surveys were conducted using the same fishing gear, it would also be important to calculate the Catch per Unit Effort and compare.
*How did authors extrapolate fish biomass per species? Did they use any surrogate per species or the calculation was direct?
*Lines 257-261, authors CANNOT conclude anything there because it is RESULTS. Conclusions go at the end of the Discussion section.
*Figures urgently require figure captioning to undestand what is depicted.
*Some graphs lack axes labels. Please, include.
*I urge authors to keep the most important figures only.
*Lines 351-353 is repetitive to what was mentioned in the Results. Please, remove.
*I strongly recommend authors to find a better way to begin with the Discussion section.
*Line 471, I am confused. Why did authors mention "fish resources"? Which fish species surveyed are of commercial importance? This was not analyzed.
English requires substantial revision.
Reviewer 2 Report
This manuscript describes the relationship between fish fauna and environmental factors around Hainan Island, south China, and may have important implications for coastal ecosystem conservation and fisheries management in the study area. However, this manuscript is not sufficient as a research article. In particular, the absence of descriptions of each figure (Figure Legends), either in the manuscript or in the appendix, is critical.
Although this article is considered to contain important findings, it is not possible to accurately evaluate the findings of the article in its current state. Therefore, I suggest that the authors revise and resubmit the manuscript.
